# Effects of Renal Denervation on the Enhanced Renal Vascular Responsiveness to Angiotensin II in High-Output Heart Failure: Angiotensin II Receptor Binding Assessment and Functional Studies in Ren-2 Transgenic Hypertensive Rats

**DOI:** 10.3390/biomedicines9121803

**Published:** 2021-11-30

**Authors:** Zuzana Honetschlägerová, Lucie Hejnová, Jiří Novotný, Aleš Marek, Luděk Červenka

**Affiliations:** 1Center for Experimental Medicine, Institute for Clinical and Experimental Medicine, 1958/9 Vídeňská, CZ-140 00 Prague, Czech Republic; 2Department of Physiology, Faculty of Science, Charles University, CZ-140 00 Prague, Czech Republic; lucie.hejnova@natur.cuni.cz (L.H.); novotnj99@natur.cuni.cz (J.N.); 3Department of Synthesis of Radiolabeled Compounds, Institute of Organic Chemistry and Biochemistry of the Czech Academy of Sciences, CZ-140 00 Prague, Czech Republic; ales.marek@uochb.cas.cz; 4Department of Pathophysiology, 2nd Faculty of Medicine, Charles University, CZ-140 00 Prague, Czech Republic; luce@ikem.cz

**Keywords:** heart failure, aorto-caval fistula, renal denervation, ANG II receptors

## Abstract

Detailed mechanism(s) of the beneficial effects of renal denervation (RDN) on the course of heart failure (HF) remain unclear. The study aimed to evaluate renal vascular responsiveness to angiotensin II (ANG II) and to characterize ANG II type 1 (AT_1_) and type 2 (AT_2_) receptors in the kidney of Ren-2 transgenic rats (TGR), a model of ANG II-dependent hypertension. HF was induced by volume overload using aorto-caval fistula (ACF). The studies were performed two weeks after RDN (three weeks after the creation of ACF), i.e., when non-denervated ACF TGR enter the decompensation phase of HF whereas those after RDN are still in the compensation phase. We found that ACF TGR showed lower renal blood flow (RBF) and its exaggerated response to intrarenal ANG II (8 ng); RDN further augmented this responsiveness. We found that all ANG II receptors in the kidney cortex were of the AT_1_ subtype. ANG II receptor binding characteristics in the renal cortex did not significantly differ between experimental groups, hence AT_1_ alterations are not responsible for renal vascular hyperresponsiveness to ANG II in ACF TGR, denervated or not. In conclusion, maintained renal AT_1_ receptor binding combined with elevated ANG II levels and renal vascular hyperresponsiveness to ANG II in ACF TGR influence renal hemodynamics and tubular reabsorption and lead to renal dysfunction in the high-output HF model. Since RDN did not attenuate the RBF decrease and enhanced renal vascular responsiveness to ANG II, the beneficial actions of RDN on HF-related mortality are probably not dominantly mediated by renal mechanism(s).

## 1. Introduction

Heart failure (HF) is a clinical syndrome which has become a major public health problem and the current treatment strategies fail to substantially attenuate the bleak prognosis, particularly in patients in whom the disease is associated with the development of renal dysfunction (“cardiorenal syndrome”) and who were originally hypertensive [1,2,3]. Therefore, new therapeutic approaches for the treatment of cardiorenal syndrome and HF are urgently needed and focused experimental studies are required to achieve this goal.

Renal denervation (RDN) involving ablation of renal afferent and efferent nerve fibers was initially developed for the treatment of resistant hypertension and, despite some initial controversies, hypertension remains the main indication for RDN [4,5,6]. The current attitude of the European Society of Hypertension is that endovascular device-based RDN in humans is a safe and efficient method with durable antihypertensive effects and should be considered as an antihypertensive treatment option that not only reduces blood pressure, but also might contribute to improved cardiovascular prognosis in hypertensive patients [7]. In addition, there is vast evidence of the importance of the sympathetic nervous system (SNS) in the regulation of renal function [8,9] and the role of SNS in the progression of HF [10,11]. This is why RDN has also been considered for the treatment of HF with promising preliminary results [12,13,14,15]. Nevertheless, an evaluation of the possible beneficial effects of RDN on long-term survival in patients with HF has not yet been conducted. The prerequisite for such clinical studies is successful experimental pre-clinical research [12,15]. Therefore, we recently explored the effects of RDN on the course of HF-dependent mortality in the model of high-output HF resulting from chronic volume overload induced by the aorto-caval fistula (ACF) in two animal models of hypertension. In the study employing the fawn-hooded hypertensive (FHH) rat, a unique genetic model of spontaneous hypertension associated with early development of chronic kidney disease (CKD), we found that RDN did not attenuate HF-dependent mortality and did not exhibit any essential renoprotective actions [16]. This finding was contradicted by our studies with Ren-2 transgenic rats (TGR), a model that displays the two most detrimental features for the progression of HF: hypertension and augmented activity of the renin-angiotensin system (RAS). We found that RDN markedly attenuated HF-dependent mortality and substantially improved long-term survival in ACF TGR [17]. Moreover, we found that renal blood flow (RBF) is decreased at a very early stage in this high-output HF model, and persistent renal dysfunction seems to be the main factor reducing the long-term survival [18,19,20,21,22]. Therefore, we hypothesized that the beneficial effects of RDN are dominantly mediated by attenuation of the development of renal dysfunction, particularly by prevention or at least retardation of RBF decrease. This interpretation has been further supported by the finding that ACF TGR exhibit selectively enhanced RBF responsiveness to angiotensin II (ANG II) [17,22], which is believed to be crucial for RBF decrease in ACF TGR. Since renal sympathetic nerve activity (RSNA) substantially modulates renal vascular reactivity to vasoactive agents, particularly in ANG II-dependent models of hypertension [23,24], the beneficial effects of RDN on RBF in ACF TGR could be mediated by abolishing, or at least by diminishing, exaggerated renal vascular responsiveness to ANG II. However, we found that in compensated HF (100% survival), RDN did not attenuate the RBF decrease and, surprisingly, it further augmented already enhanced RBF responsiveness in ACF TGR [17]. In addition, we found that the kidney ANG II type 1 (AT_1_) receptor protein levels did not significantly differ between intact (non-denervated) sham-operated TGR and intact ACF TGR, and RDN did not alter them. A limitation of our earlier study was that we did not evaluate AT_1_ and ANG II type 2 (AT_2_) receptor density and affinity in the kidney [17]. Therefore, the present study examined renal vascular responsiveness to ANG II along with characterization of AT_1_ and AT_2_ receptors in the kidney by ligand binding studies in ACF TGR that were or were not subjected to RDN. In the present study, we focused on the characterization of the role of AT_2_ receptors. Increasing evidence suggests that their activation not only counteracts AT_1_ receptor-mediated vasoconstriction [25,26], but can also exhibit organ-protective actions under various pathophysiological circumstances [25,26,27]. The assessment of the vascular responsiveness to ANG II was conducted two weeks after RDN, i.e., three weeks after the creation of the ACF. At that time, the non-denervated ACF TGR displayed the onset of HF decompensation (15–20% mortality rate) whereas the post-RDN rats showed a 100% survival rate. The aim of the present study was to obtain a deeper insight in the role of exaggerated RBF responsiveness to ANG II in the onset of HF decompensation in a reliable model of severe high-output HF. Our previous results were not fully anticipated; therefore, in order to confirm them we evaluated RBF responsiveness to ANG II in separate groups of animals. Moreover, we aimed to make sure that the biochemical characteristics of kidney ANG II receptors accord with our physiological findings; therefore, we performed the relevant studies in the same animal groups.

## 2. Materials and Methods

### 2.1. Animals

All animals used in the present study were bred at the Center of Experimental Medicine of this Institute, from stock animals supplied by the Max Delbrück Center for Molecular Medicine, Berlin, Germany, which is accredited by the Czech Association for Accreditation of Laboratory Animal Care. Heterozygous TGR were generated by breeding male homozygous TGR with female homozygous Hannover Sprague-Dawley (HanSD) rats, and age-matched HanSD rats served as transgene-negative normotensive controls. The animals were kept on a 12 h/12 h light/dark cycle. Throughout the experiments rats were fed a normal-salt, normal-protein diet (0.45% NaCl, 19–21% protein) and had free access to tap water. Male TGR rats at the initial age of 8 weeks were used for experiments: at this age, TGR are already in the sustained phase of hypertension with blood pressure (BP) levels comparable with those of hypertensive patients [systolic BP (SBP) about 180 mmHg] and with substantial activation of endogenous RAS, as demonstrated in previous studies including ours. Therefore, the animals seemed to be a suitable model for studying the role of hypertension and endogenous RAS in the pathophysiology of HF [18,19,22,28]. 

### 2.2. Heart Failure Model, Exclusion Criteria, and RDN Technique

Eight-weeks-old male TGR rats were anesthetized with an intraperitoneal injection of ketamine/midazolam mixture (Calypsol, Gedeon Richter, Hungary, 160 mg/kg and Dormicum, Roche, France, 160 mg/kg). Chronic HF due to volume overload was then induced by creating an ACF using a needle technique. This procedure is routinely performed in our laboratory and the details of the technique were reported previously [16,17,18,19,20,21,22,28]. Sham-operated rats underwent an identical procedure but without creating ACF. A standard bilateral RDN or sham-RDN procedure was performed one week after ACF or sham surgery. Under ketamine/midazolam anesthesia, both kidneys were approached through a midline abdominal incision, both renal arteries and veins were stripped of the connective tissue, including the perivascular adipose tissue that was carefully removed, and all the visible nerve fibers were separated from the arteries and cut close to the hilus. Then, both renal arteries were painted with a 10% phenol solution in absolute alcohol in order to destroy any remaining nerve fibers. During the application of phenol, the adjacent tissues were carefully protected from exposure to the phenol solution and disruption of the major lymphatic vessels in the area was avoided. In case a sustained renal artery spasm (longer than 10 s) was observed, the animal was excluded from the study. This was done to minimize the risk of phenol-induced damage of the renal artery (such as stenosis or endothelial damage that in the long-term could cause intrarenal thrombosis). In total, three animals from the sham-operated group and five animals from the ACF group were excluded from the study. Such exclusion criteria were also used in our recent studies employing RDN in rats [16,17]. Control animals underwent laparotomy and retraction of the abdominal organs, but renal vessels were not isolated or painted with phenol solution and the renal nerves were not severed; instead, the intraabdominal area was coated with a 0.9% NaCl solution. This procedure of RDN which causes ablation of both afferent and efferent renal nerves is now a golden standard method of RDN, and was repeatedly shown by many research groups, including ours, to be fully effective [8,9,12,16,17,24,28,29,30,31].

### 2.3. Detailed Experimental Design

#### 2.3.1. Series 1: Effects of RDN on RBF Responsiveness to ANG II

The animals underwent either sham-operation or ACF creation as described above (on the day labeled −7) and on day 0 they were randomly assigned to the group that was exposed to RDN or to the sham procedure. Two weeks later, the rats were anesthetized with thiopental sodium (80 mg/kg i.p.) and prepared for mean arterial pressure (MAP) and RBF measurement as well as for renal artery injections of the agents tested, as described in detail in our recent publications [17,20,21,22]. Briefly, the right jugular vein was catheterized with polyethylene (PE50) tubing for infusion of solutions and intravenous drug administration. The right femoral artery was cannulated for continuous monitoring of arterial blood pressure. MAP was monitored using a pressure transducer and recorded using a computerized data acquisition system (PowerLab, ADInstruments, Oxford, UK). The left kidney was exposed via a flank incision, isolated from the surrounding tissue, and placed in a lucite cup, and the ureter was cannulated with a PE-10 catheter. For selective intrarenal administration, a tapered PE-10 catheter was inserted into the aorta via the left femoral artery and passed 1–2 mm down the left renal artery. This catheter was kept patent by continuous infusion of heparinized isotonic saline at 2 µL/min throughout the experiment. During the surgery, animals received an intravenous infusion of 0.9% saline solution containing 6% bovine serum albumin (Sigma Chemical Co., Prague, Czech Republic) at a rate of 20 µL/min. With the surgery completed, an isotonic saline solution was infused to compensate for fluid losses. An ultrasonic transient-time flow probe (1RB, Transonic Systems, Altron Medical Electronic GmbH, Fürstenfeldbruck, Germany), connected to a Transonic flowmeter, was placed on the left renal artery and RBF was continuously recorded. After the completion of surgery, a 45-min equilibration period was allowed and then one 30-min period was made, to determine initial MAP and basal RBF. Subsequently, renal vascular responses to intrarenal bolus of ANG II (8 ng) were determined. We used a similar experimental protocol as in the recent study [17] to confirm the pattern of renal vascular responses to ANG II before performing the characterization of AT_1_ and AT_2_ receptors using ligand binding studies. Briefly, ANG II was loaded in a small volume (20 µL) to the Cheminert valve (Harvard apparatus) and then rapidly infused to the animal as a bolus in saline solution (100 µL). To deliver the entire dose of ANG II into the kidney the rate of infusion through the renal artery catheter was increased for 30 s from 2 to 100 µL/min. 

Analogously, MAP and RBF responses to the intrarenal arterial bolus of 100 µL of isotonic saline solvent were evaluated in order to exclude the effects of intrarenal volume administration *per se*. The order of the administration of ANG II or solvent was randomized. The baseline values of MAP and RBF were always assessed separately for each administration. The renal vascular resistance (RVR) was calculated by the standard formula. This method of assessment of renal vascular reactivity in vivo was originally introduced by the Arendshorst group [32] and then modified for use in TGR in Mitchell’s laboratory [33], and has been routinely employed in our laboratory. It enables a comparative assessment of the actions of various active agents on renal vascular responsiveness in various models of diseases, such as experimental hypertension and HF [20,21,22,34,35].

At the end, the left ventricle (LV) weight (including septum), right ventricle (RV) weight, and lung weight were assessed. In the case of the lungs, the so called “wet lung weight” was assessed, i.e., the excised lung was only gently pressed and left for five minutes on the suction filter paper. This method is consistently used in our laboratory and enables the comparison of the effects of various treatment regimens on the degree of cardiac hypertrophy and lung congestion in the ACF model of HF [16,17,18,19,20,21,22,28].

The following experimental groups were examined (*n* = 10 in each group):(a)Sham-operated TGR (no ACF) + sham RDN(b)Sham-operated TGR + RDN(c)ACF TGR + sham RDN(d)ACF TGR + RDN

#### 2.3.2. Series 2: Effects of RDN on Kidney ANG II Receptor Characterization Using Ligand Binding Studies

Animals were prepared as described in series 1, (actually, at the same time) and on the day of the experiment were randomly assigned either to series 1 or to series 2. Again, on day 0 either the sham-operation or RDN was performed and two weeks later animals were euthanized by decapitation and the same experimental groups (again: *n* = 10 in each group) were evaluated as in the first series. The kidneys were dissected longitudinally, the medulla and papilla were discarded, and the kidney cortex tissue samples were immediately harvested into liquid nitrogen and stored at −80 °C until the analysis.

### 2.4. Preparation of Crude Membrane Fraction

On the day of the analysis, the frozen cortexes of the left and right kidney (300 mg each) from all animals (10 rats per group) were assembled and the fivefold volume of ice-cold TMES buffer (20 mM TRIS, 3 mM MgCl_2_, 1 mM EDTA, 250 mM sucrose; pH 7.4) containing protease and phosphatase inhibitors (cOmplete, Sigma-Aldrich, St. Louis, MO, USA) was added. Samples were homogenized for 15 sec using Ultra-Turrax (24,000 rpm) and then for 1 min in a glass homogenizer with a motor-driven Teflon pestle at 1200 rpm. All homogenization steps were undertaken on ice. Homogenates were centrifuged at 800× *g* for 10 min at 4 °C (Hettich Universal 320 R centrifuge), the supernatants were collected and the pellet re-homogenized in TMES and centrifuged again under the same conditions. Both supernatants were mixed and centrifuged at 50,000× *g* for 30 min at 4 °C (Beckman Optima Max L90K ultracentrifuge). The pellets were suspended in the TME buffer (20 mM TRIS, 3 mM MgCl_2_, 1 mM EDTA; pH 7.4) and re-centrifuged. The resulting pellets were suspended in the TME buffer and crude membrane fractions were aliquoted and stored at −80 °C until use. Protein concentration was measured using a BCA protein assay.

### 2.5. ANG II Receptors Binding Assay 

The total amount of ANG II receptors in crude membrane fractions prepared from rat renal cortex was determined using homogeneous competition radioligand binding assays that were employed previously by various research groups [36,37,38,39,40,41] and were modified in our laboratories. Briefly, samples of crude membrane fraction (80 μg protein) were incubated in 0.2 mL of the incubation buffer (50 mM Tris-Cl, 10 mM MgCl_2_, 100 mM NaCl and 0.25% BSA; pH 7.4) with 2 nM of the radiolabeled ANG II analogue, [^125^I][Sar1,Ile8]AngII, and growing concentration of [Sar1,Ile8]ANG II (10^−11^–10^−6^ M) at 25 °C for 1 h. The binding reaction was terminated by adding 3 mL of ice-cold incubation buffer and subsequent filtration through GF/C filters presoaked for 1 h in the buffer (50 mM Tris-Cl, 10 mM MgCl_2_, 100 mM NaCl and 1% BSA; pH 7.4). The filters were then washed 3 times with 3 mL of the ice-cold incubation buffer. Radioactivity retained on the filters was measured by liquid scintillation counting in 4 mL scintillation cocktail EcoLite for 5 min. Binding characteristics, i.e., the maximal density (B_max_) and affinity (K_D_) of ANG II receptors were calculated using GraphPad Prism 7 software (GraphPad Software, San Diego, CA, USA). AT_2_ subtype was determined using selective antagonist PD 123319 in one concentration. Crude membranes were incubated with 2 nM [^125^I][Sar1,Ile8]ANGII without or with 10^−5^ M PD123319 under the same conditions as described above. Nonspecific binding was determined by incubation in the presence of 1 μM ANG II. The AT_1_ receptor binding was defined as the difference between nonspecific binding minus binding in the presence of PD 123319, the AT_2_ receptor blocker. The detailed procedure of preparation of the radiolabeled ANG II analogue, [^125^I][Sar1,Ile8]AngII is described in the Appendix A.

### 2.6. Statistical Analysis

A statistical analysis of the data was performed using Graph-Pad Prism 7 (Graph Pad Software, San Diego, CA, USA). The statistical comparison was made by a one-way ANOVA when appropriate. The values were expressed as mean ± SEM; *p* < 0.05 was considered to indicate a statistically significant difference.

### 2.7. Statement of Ethics

The study followed the guidelines and practices established by the Animal Care and Use Committee of IKEM, which accord the national laws and are in accordance with the recommendation of the European Union. The protocol of the study was approved by the Animal Care and Use Committee of the IKEM and, subsequently, by the Ministry of Health of the Czech Republic (project decision on 15 April 2019, 14553/2019-4/OVZ).

## 3. Results

### 3.1. Series 1: Effects of RDN on Renal Vascular Responsiveness to ANG II

Figure 1A–C shows the collected basal MAP, RBF, and RVR values measured in rats 3 weeks after the creation of ACF (or sham-operation) and 2 weeks after RDN (or in intact, i.e., non-denervated animals). The creation of the ACF caused a significant decrease in MAP and this was not modified by RDN. On the contrary, in sham-operated TGR (no ACF), RDN elicited a significant decrease in MAP as compared with non-denervated sham-operated TGR (Figure 1A). 

The basal RBF was markedly lower in intact ACF TGR as compared with intact sham-operated TGR (about 35% reduction) and RDN did not change the basal RBF, similarly in sham-operated and ACF TGR (Figure 1B). As shown in Figure 1C, RDN significantly decreased RVR in sham-operated TGR as compared with intact sham-operated TGR. Intact ACF TGR did not exhibit higher RVR as compared with intact sham-operated TGR, but denervated ACF TGR displayed higher RVR compared with denervated sham-operated TGR (Figure 1C).

As shown in Figure 1D, the intrarenal bolus administration of ANG II (8 ng) did not alter MAP in either group and the intrarenal bolus administration of isotonic saline solution of the same volume elicited the same changes in MAP as seen with the intrarenal bolus of ANG II, thus, there was no change in MAP (data not shown).

Figure 1E,F present the maximum decreases in RBF elicited by intrarenal ANG II bolus (8 ng) expressed either in absolute values (Figure 1E) or as percent changes (Figure 1F). Evidently, RBF decreases were substantially larger in non-denervated ACF TGR compared with intact sham-operated TGR. The RBF decreases produced by intrarenal ANG II were more profound in post-RDN than in intact animals, and this was clearly seen in sham-operated TGR as well as in ACF TGR (Figure 1E,F).

Figure 2 collects the organ weight data and the effects of RDN. Intact ACF TGR showed a striking bilateral cardiac hypertrophy as evident from the LV and RV weights (27% and 107% increases, respectively). RDN did not modify LV or RV weight in sham-operated TGR (no ACF), but it significantly decreased them in ACF TGR; however, LV and RV weights remained higher than in sham-operated TGR (Figure 2A,B).

The wet lung weight was employed as a marker of lung congestion, as is generally practiced in this model of high-output HF [16,17,18,19,20,21,22,28,42,43]; the value was substantially increased (about 41%) in non-denervated ACF TGR compared with intact sham-operated TGR. In sham-operated TGR, RDN did not modify lung weight but reduced it significantly in ACF TGR; however, the values still remained significantly higher (about 11%) than in sham-operated TGR.

### 3.2. Series 2: Effects of RDN on Kidney ANG II Receptors Characterization Using Ligand Binding Studies

As summarized in Table 1, the total number and affinity of ANG II receptors in crude membrane from the renal cortex did not differ between experimental groups. Representative binding curves of the homogenous competition binding experiments in individual experimental groups are presented in Figure 3. As shown in Table 2, the selective antagonist of AT_2_ receptors did not inhibit, to any extent, the binding with radiolabeled ANG II analogue ([^125^I][Sar1,Ile8]ANG II) in any of the experimental groups, indicating that AT_2_ receptors were not present in rat crude membranes isolated from the renal cortex. This indicates that all the ANG II receptors detected are of the AT_1_ receptor subtype.

## 4. Discussion

Our present study aimed to evaluate the hypothesis that exaggerated RBF responsiveness to ANG II at the onset of the decompensation phase in the high-output HF model (i.e., in ACF TGR) is mediated by alterations of the ANG II receptor binding characteristics. We assumed that the increased vascular responsiveness, possibly further augmented by RDN, depends on (1) an increase in density and/or affinity of AT_1_ receptors, which are dominantly responsible for ANG II-mediated renal vasoconstriction [44,45], or (2) a decrease in density and/or affinity of AT_2_ receptors, whose activation is suggested to counteract AT_1_ receptor-mediated vasoconstriction [25,26]. We focused on the role of AT_2_ receptors, in consideration of the growing recognition that their activation could *per se* have wide-ranging protective effects [25,26,27]. Therefore, in the present study we performed binding studies in the absence and in the presence of a highly selective AT_2_ receptor antagonist. We thought it important to focus on the AT_1_ and AT_2_ receptors since comprehensive evaluation of renin, angiotensin-converting enzyme, AT_1_ and AT_2_ receptor mRNA and protein expressions clearly demonstrated local activation of the RAS in the heart and kidneys in the ACF model of HF [46]. Moreover, the magnitude of RAS activation in these tissues was proportional to the severity of HF [46]. However, regarding the AT_1_ and AT_2_ receptor expression, the quoted study did not reach unequivocal conclusions: the authors speculated that a shift in ANG II receptor subtypes, i.e., AT_1_ downregulation and AT_2_ upregulation, could be a local protective mechanism in this HF model.

The first important finding of the present study is that the AT_2_ receptor binding was not detectable in the renal cortex of any experimental group; therefore, it is apparent that all operating ANG II receptors are of the AT_1_ subtype. The results of previous studies in various experimental models (characterized by augmentation of intrarenal ANG II concentrations), such as ANG II-infused hypertensive rats [36,40,41,47], two-kidney, one-clip (2K1C) Goldblatt hypertensive rats [37], and rats with HF induced by myocardial infarction (MI) [39] indirectly support our present concept. In each of these models, including our ACF TGR model of HF, ANG II receptors in the kidney were exclusively of the AT_1_ subtype. However, it has to be admitted that in none of the studies, including ours, were other ANG II receptor types, such as type 4 or Mas receptors for angiotensin-1-7, evaluated. Thus, the present findings indicate that alterations of the AT_2_ receptors do not play any important role in the exaggerated RBF responsiveness to ANG II in ACF TGR that were or were not subjected to RDN. This conclusion is also supported by our previous studies employing genetically engineered mice [48,49,50]. It was shown that acute or chronic administration of a selective AT_2_ receptor blocker or selective AT_2_ receptor stimulator did not alter circulating and intrarenal concentrations of ANG II and, in particular, did not modify the course of hypertension in 2K1C Goldblatt animals. This suggests that AT_2_ receptors do not play any important role in the regulation of ANG II production and blood pressure in this model of ANG II-dependent hypertension. However, an ultimate conclusion about the lack of the role of AT_2_ receptors in this high-output HF model requires studies involving the chronic application of selective AT_2_ receptor blockers or selective AT_2_ receptor agonists. 

Remarkably, our finding that in the non-denervated ACF TGR the total number of ANG II receptors did not increase in the renal cortex and this was not altered by RDN is in disagreement with the observations made by Clayton et al. [51] in rabbits with HF induced by sustained ventricular pacing. The authors reported that the development of HF in this model was accompanied by increasing AT_1_ and decreasing AT_2_ receptor protein expression in the renal cortex. Moreover, nonselective RDN substantially attenuated these changes in rabbits with chronic HF, and ANG II receptor expression did not change in healthy controls. Importantly, the positive effect of RDN in HF rabbits was more pronounced in the case of changes of the AT_2_ receptors: the authors postulated that modulation of the renal AT_2_ receptors might be crucial for the preservation of RBF in the setting of chronic HF [51]. There is no satisfactory explanation for the discrepancy between these findings and our results. Evidently, Clayton et al. [51] employed a different model of HF (pacing-induced HF in rabbits) and, probably less important, a different method of ANG II receptor assessment in the renal cortex (protein expression by Western methodology).

The second set of findings relates to the AT_1_ receptor binding characteristics because it is thought that vascular and glomerular ANG II receptors in the kidney are downregulated during the conditions that raise circulating and intrarenal ANG II concentrations, and upregulated under opposite conditions [44,52]. However, this largely accepted and in most cases valid notion is an oversimplification. It has been shown that while alterations in intrarenal ANG II levels by a low-sodium diet resulted in expected downregulation of AT_1_ receptors in the preglomerular vasculature [36], in the early phase of 2K1C hypertension the elevated ANG II levels did not induce the anticipated downregulation of vascular AT_1_ receptors [37]. In addition, the quoted study showed different regulation of glomerular and preglomerular vascular AT_1_ receptors: they were unchanged under the chronic low-sodium diet [36] but downregulated in 2K1C hypertensive rats [37]. It was also reported that the AT_1_ receptor density and affinity were not altered in the whole kidney of rats studied 4–6 weeks after MI while, surprisingly, the glomerular AT_1_ receptors were markedly upregulated. This was so despite the fact that the intrarenal RAS was prominently activated [39]. Furthermore, it was shown that AT_1_ receptor binding was maintained in the renal cortex of ANG II-infused hypertensive rats, in spite of augmented intrarenal ANG II concentrations [40,41]. We found no significant between-group differences in ANG II receptor binding characteristics in the renal cortex, which indicates that in ACF TGR the exaggerated RBF responsiveness to ANG II cannot be simply ascribed to increases in the number and/or affinity of AT_1_ receptors, similarly without or after RDN. In this context, it is important to emphasize that all our studies in ACF TGR [18,22], including the most recent ones [17,28], have clearly demonstrated that kidney ANG II concentrations are at this stage of HF markedly elevated compared with sham-operated TGR. In addition, in our previous study we found that RDN did not alter renal concentrations of ANG II in ACF TGR at this stage of HF [17]. 

Moreover, in our most recent study in FHH rats, we confirmed that ACF creation was followed by marked increases in kidney ANG II concentration and, again, this was not altered by RDN [16]. Evidently, RDN did not directly alter intrarenal RAS activity, similarly in ACF TGR and ACF FHH [16,17]. Therefore, in the high-output model of HF the beneficial effects of RDN cannot be simply ascribed to the suppression of the vasoconstrictor/sodium retaining axis of the RAS. However, even though our findings do not explain the selectively exaggerated RBF responsiveness to ANG II in ACF TGR, the lack of kidney AT_1_ receptor downregulation in the presence of augmented intrarenal ANG II is very striking: it provides the basis for the sustained influence of the intrarenal RAS on the renal hemodynamics and tubular reabsorption, and consequently on the development of renal functional derangements in ACF TGR. Therefore, our present findings, along with our recent evidence that a selective intrarenal AT_1_ receptor blockade elicits greater increases in RBF than in intact sham-operated TGR [17], are consistent with the concept that in the high-output HF model renal functional derangements are strongly ANG II-dependent. It is noteworthy that in our previous studies we found that ACF TGR exhibited exaggerated renal vasoconstrictor responses to ANG II whereas intrarenal administration of NE elicited in ACF TGR an RBF decrease comparable to that observed in sham-operated TGR [17,22]. Thus, in ANG II-dependent hypertension the high-output HF rats displayed a renal vasoconstrictor hypersensitivity to ANG II, but not to NE. Admittedly, we did not determine the effects of selective intrarenal α-adrenergic blockade (e.g., using phentolamine) on the renal hemodynamics and renal excretory function in intact ACF TGR, and did not compare the effects with those in intact sham-operated TGR. Therefore, we cannot fully rule out the possibility that even though ACF TGR do not exhibit exaggerated renal vascular responsiveness to NE, its augmented intrarenal levels [22,53] have some persistent adverse effects on the renal hemodynamics and tubular reabsorption. Appropriate studies are obviously needed.

The third important issue of our study relates to the long-term physiological and pathophysiological importance of the further augmentation of RBF responsiveness to ANG II after RDN. Our observation resembles the well-known phenomenon of post-denervation supersensitivity to NE, the result of the loss of presynaptic neuronal uptake sites for NE [54]. The initial acute in vivo experiments suggested that the supersensitivity of the renal vasculature and tubules to NE could in the long-term annihilate the beneficial actions of RDN, particularly under conditions where NE levels are elevated [30,31,55,56]. Later chronic studies in conscious animals clearly demonstrated that such an untoward effect is unlikely [57,58]. Although the mechanism underlying RBF supersensitivity to ANG II after RDN is probably different from that determining supersensitivity to NE, and unclear, it does not counteract the beneficial actions of RDN, as is apparent from the substantial attenuation of HF-dependent mortality in ACF TGR that underwent RDN [17]. Nevertheless, it is obvious that further studies are needed to evaluate the mechanism(s) underlying renal vascular supersensitivity to ANG II in our sham-operated and, in particular, our ACF TGR.

The fourth important issue is that of the actual mechanism(s) underlying the beneficial effects of RDN on HF-related mortality and organ morphology in ACF TGR as shown in our recent [17] and also in the present study. The original hypothesis was that the development of renal dysfunction, as seen from the reduction in RBF, is at least in part mediated by the inappropriate activation of the renal sympathetic efferent nerves, and that the augmented activation of the “Brain-Kidney Neural Axis”, as recently formulated by Osborn [9], importantly contributes to the reduction of RBF in HF [9,12]. Therefore, RDN obtained by our technique (ablation of both afferent and efferent renal nerves) should prevent or at least attenuate the decreases in RBF in the high-output HF model. However, our previous [17] and present data have convincingly demonstrated that the RDN did not attenuate, to any degree, the reduced RBF in ACF TGR. Therefore, it is tempting to propose that the beneficial actions of RDN are not mediated by renal mechanism(s). However, two recent studies, in ovine HF model and in sheep with hypertensive CKD, showed that RDN not only lowered blood pressure but also improved the glomerular filtration rate (GFR) and sodium and water excretion [59,60]. In addition, RDN was recently reported to attenuate the increased expression of renal epithelial sodium channels (ENaC) and the water channel aquaporin 2 (AQP2) in rats with MI-induced HF [61]. Noteworthily, increased renal expression of ENaC and AQP2 has been implicated in the renal sodium and water retention in HF [61,62,63]. Such studies strongly suggest that RDN may help improve kidney function, thus, in HF individuals RDN may be beneficial for the heart as well as for the kidneys. Nevertheless, to conclusively resolve the issue if the beneficial actions of RDN on the long-term course of HF in ACF TGR are or are not mediated by the improvement of renal function, further comprehensive studies evaluating not only RBF, but also GFR and renal tubular functions (e.g., renal micropuncture studies) are necessary [64]. 

Are the beneficial actions of RDN mediated by cardiac mechanism(s)? In both our studies, RDN, applied at the onset of the HF decompensation phase, attenuated cardiac hypertrophy and lung congestion. This suggests that, in fact, the beneficial actions of RDN in ACF TGR might be mediated by the improvement of the cardiac function. However, to resolve this question, a more detailed evaluation of the cardiac structure and function is needed, including echocardiography and an invasive pressure-volume analysis. In this context, it should be emphasized that the kidneys are also innervated by sensory (afferent) nerves that relay information to the brain and modulate sympathetic outflow to many organs. It has been proposed that enhanced activation of the “Kidney-Brain Neural Axis”, as defined by Osborn [9], might be implicated in the pathophysiology of many cardiovascular diseases, including HF [9,12,65]. Therefore, the beneficial actions of our RDN might depend on the ablation of afferent renal nerves, which could explain the lack of actions of our RDN technique on the RBF in ACF TGR. This notion is supported by recent findings showing that selective targeting of the “Kidney-Brain Neural Axis” substantially attenuated hypertension and end-organ damage in 2K1C hypertensive rats and in the rabbit model of CKD [66,67]. Therefore, to test our new hypothesis it is necessary to examine the effects of selective ablation of renal sensory nerves on the course of HF in ACF TGR, e.g., using periaxonal capsaicin administration, a method developed originally by Foss et al. [68]. However, it should be cautioned that the method has an important limitation: it is effective over 2 weeks only [66,68]; thereafter, considerable reinnervation occurs, which limits, even though it does not exclude, application in long-term studies that are needed to assess the animals´ survival. 

The fifth interesting finding is that RDN performed by our method (ablation of both afferent and efferent renal nerves) significantly reduced MAP in sham-operated TGR but not in ACF TGR. Based on our present data we cannot provide a plausible explanation for such divergent actions of RDN. To address this issue, studies employing the radiotelemetric blood pressure measurement in conscious animals are needed.

## 5. Limitations of the Study

The first limitation of our study is that ANG II receptor binding was examined in homogenates of the whole renal cortex. Some studies suggest that glomerular, vascular, and also tubular AT1 receptors respond to the changes of intrarenal ANG II levels differently, e.g., in the opposite direction [36,37,39,69,70]. Because of the unsuccessful attempts to isolate renal vasculature we decided to employ the technique previously used for the evaluation of ANG II receptor binding characteristics in the ANG II-infused hypertensive rats [40,41]: a crude membrane fraction from the homogenized renal cortex was used. Future studies (classical in vitro autoradiography, immunohistochemistry, and RT-PCR techniques) will be needed to precisely evaluate the location and distribution of ANG II receptors within the kidney, particularly in the vasculature [71,72,73].

The second limitation relates to the use of sodium thiopental, an anesthetic that might alter RSNA [8,9,24] which was not measured in this study. While renal NE concentration is a reliable and robust indicator of RSNA and a marker of the effectiveness and durability of the RDN, RSNA should be measured in samples obtained from conscious decapitated animals [74,75]. Regrettably, we had to evaluate the renal vascular responsiveness to vasoactive agents under deep anesthesia and RSNA should be measured in conscious animals. Thus, our assumption of increased RSNA in intact and decreased RSNA in denervated sham-operated, as well as in ACF TGR, is based solely on our previous data on renal NE concentration.

The third limitation relates to the hypertension model employed. The transgenic rat line [official strain name: TGR(mRen2)27] is the first successful transgenic hypertensive rat model [76]. It is constructed by insertion of the mouse Ren-2 renin gene, including flanking sequences, into the genome of the Hannover Sprague-Dawley (HanSD) rat [76]. Although the hypertension that occurs in TGR is undoubtedly related to the insertion of the mouse Ren-2 renin gene (which in theory is a clearly monogenetic model of hypertension), the pathophysiological mechanisms responsible for the development of hypertension have not been fully elucidated [77,78,79], and many investigators consider it “non-natural”. However, despite some uncertainties, more recent studies from different laboratories, including our own, showed that TGR exhibit increased circulating and kidney ANG II concentrations and represent a clearly ANG II-dependent model of hypertension [34,74,75,80,81,82,83]. Based on the current knowledge, it displays the features of primarily RAS-dependent hypertension, and its advantage is that two crucial factors promoting the progression of HF, i.e., hypertension and increased RAS activity, are present before the induction of HF [84,85]. Evidently, future studies using different hypertensive as well as normotensive rat strains are needed to further support our conclusions.

The fourth limitation relates to the use of the ACF-induced model of HF. ACF rats represent a well-defined model of volume overload-dependent HF, characterized by the activation of systemic and intrarenal RAS, signs of bilateral cardiac remodeling, congestion, as well as the development of renal dysfunction [18,19,20,21,22,28,42,43,53]. Even though the initial insult is different from that in the majority of patients with HF, the model has many features in common with untreated human HF. Nevertheless, the model also has at least two important limitations. First, HF induced by volume overload represents only 5–7% of all HF cases, mostly patients with mitral insufficiency [86]. Second, ACF animals have high renal venous pressure [87], which may impair renal function, e.g., decrease GFR even with normal RBF [88,89]. Nevertheless, the model is recommended by the American Heart Association and the European Society of Cardiology for studying the pathophysiology of HF [90,91], particularly for evaluation of cardio-renal interactions (“cardiorenal syndrome”) [1].

Nonetheless, even considering all the aforementioned limitations, we are convinced that the results of the current studies provide important information on the pathophysiology of cardiorenal syndrome.

## 6. Conclusions

Our data show that exaggerated RBF responsiveness to ANG II in ACF TGR and its further augmentation after RDN cannot be simply explained by the increases in the number and/or affinity of renal AT_1_ receptors. The maintained kidney AT_1_ receptor binding combined with elevated ANG II levels provide the basis for the sustained influence of the RAS on the renal hemodynamics and tubular reabsorption, and consequently for the development of renal dysfunction (“cardiorenal syndrome”) in our high-output HF model. Since RDN did not attenuate the RBF decrease in ACF TGR, the beneficial actions of renal denervation on HF-related mortality are not dominantly mediated via renal mechanism(s). Therefore, further studies are needed to elucidate the mechanism(s) responsible for the favorable effects of RDN on HF-related mortality and morbidity. 

## Figures and Tables

**Figure 1 biomedicines-09-01803-f001:**
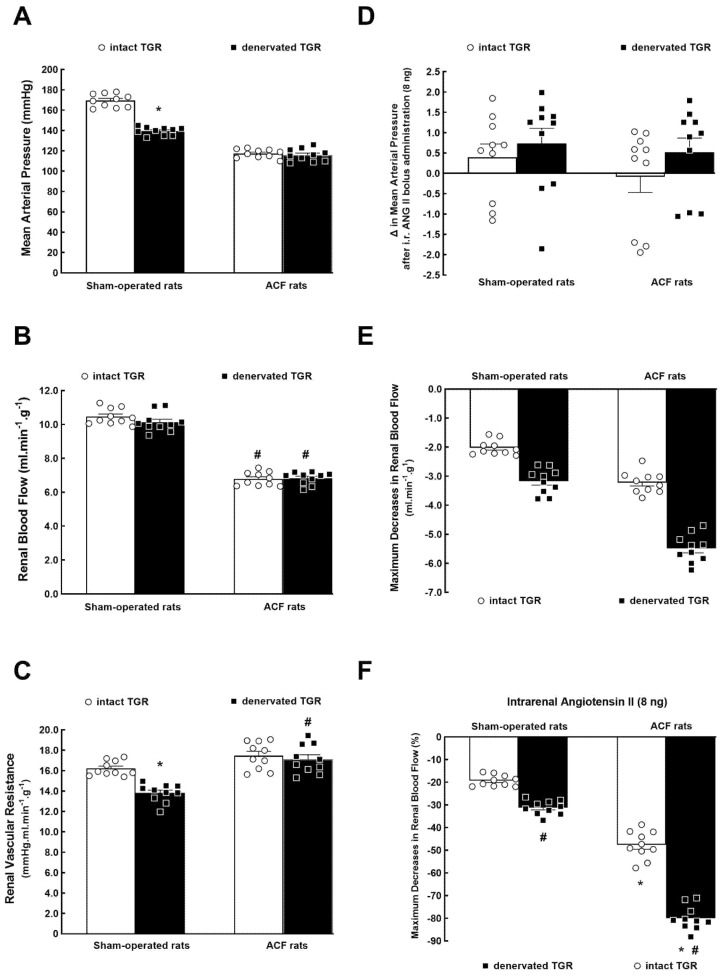
Mean arterial pressure (**A**), renal blood flow (**B**), renal vascular resistance (**C**), changes in mean arterial pressure after intrarenal bolus administration of 8 ng of angiotensin II (ANG II) (**D**), maximum decrease in RBF in absolute values (**E**) and as percent changes (**F**) induced by intrarenal bolus of 8 ng of ANG II in intact (no RDN))—blank bars) and in denervated (black bars) Ren2-transgenic hypertensive rats (TGR), either with aorto-caval fistula (ACF) or sham-operated. * *p* < 0.05 versus intact TGR. **^#^**
*p* < 0.05 versus sham-operated TGR counterparts for (**A**–**C**). * *p* < 0.05 versus sham-operated rats. **^#^**
*p* < 0.05 versus intact counterparts for (**E**,**F**). All the values shown are means ± SEM; the dots for individual values are also shown.

**Figure 2 biomedicines-09-01803-f002:**
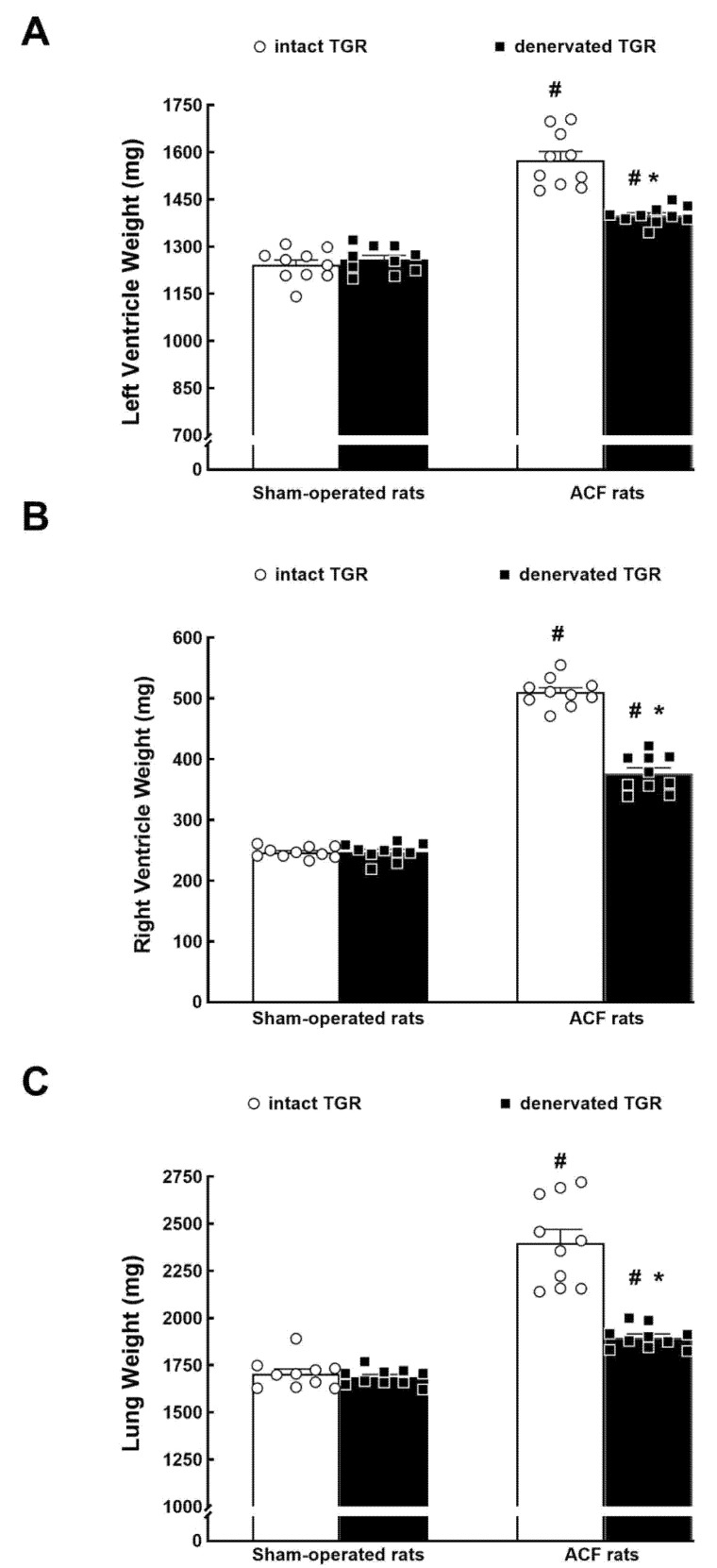
Left ventricle weight (**A**), right ventricle weight (**B**), and lung weight (**C**) in intact (no RDN—blank bars) and in denervated (black bars) Ren2-transgenic hypertensive rats (TGR) that were either with aorto-caval fistula (ACF) (right hand side) or sham-operated (left). * *p* < 0.05 versus intact TGR. **^#^**
*p* < 0.05 versus sham-operated TGR. The values are means ± SEM; the dots for individual values are also shown.

**Figure 3 biomedicines-09-01803-f003:**
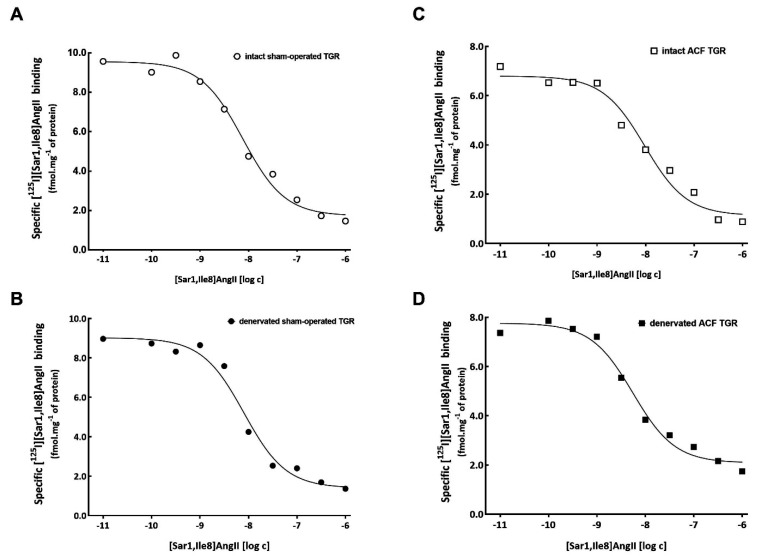
Representative binding curves of homogeneous competition binding experiments in intact, i.e., non-denervated sham-operated Ren2-transgenic hypertensive rats (TGR) (**A**), denervated sham-operated TGR (**B**), intact TGR with aorto-caval fistula (ACF) (**C**), and denervated ACF TGR (**D**). Number of binding sites (B_max_) and the dissociation constant (K_D_) of angiotensin II receptors were determined using homogeneous competition radioligand binding assays with labeled [^125^I][Sar1,Ile8]AngII and unlabeled [Sar1,Ile8]AngII. Each graph represents a typical experiment performed in triplicate.

**Table 1 biomedicines-09-01803-t001:** Binding characteristics of angiotensin II receptors in crude membrane isolated from kidney cortex.

	Sham-Operated TGR + Intact	Sham-Operated TGR + RDN	ACF TGR + Intact	ACF TGR + RDN
B_max_ (fmol mg^−1^ of protein)	22 ± 4	19 ± 3	18 ± 3	17 ± 4
K_D_ (nM)	3.5 ± 0.8	4.2 ± 0.9	3.2 ± 0.9	3.9 ± 0.8

Values are means ± SEM from three experiments performed in triplicates. B_max_, maximal binding capacity; K_D_, equilibrium dissociation constant.

**Table 2 biomedicines-09-01803-t002:** Specific [^125^I][Sar1,Ile8]Ang II binding after adding PD 123319 (a specific AT_2_ receptor antagonist for ANG II).

	Sham-Operated TGR + Intact	Sham-Operated TGR + RDN	ACF TGR + Intact	ACF TGR + RDN
% of specific [^125^I][Sar1,Ile8]Ang II binding	98 ± 7	93 ± 5	103 ± 11	92 ± 4

Values are means ± SEM from three experiments performed in pentaplicates. Results are presented as percentages of specific [^125^I][Sar1,Ile8]Ang II binding determined in the presence of 10^−5^ M PD123319 relative to specific [^125^I][Sar1,Ile8]Ang II binding in the absence of PD123319, which represents 100% value.

## Data Availability

Data contained within the article and the original data that support the findings of the present study are available from the corresponding author upon reasonable request.

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
