# Peer review of "Effects of Renal Denervation on the Enhanced Renal Vascular Responsiveness to Angiotensin II in High-Output Heart Failure: Angiotensin II Receptor Binding Assessment and Functional Studies in Ren-2 Transgenic Hypertensive Rats"

_biomedicines, 2021, doi:10.3390/biomedicines9121803_

Round 1
Reviewer 1 Report
These authors have examined the effects of surgical renal denervation on the enhanced renal vascular response to Ang II in a rat model of heart failure (Ren-2 transgenic rats with an aorto-caval fistula). Studies were performed 3 weeks after induction of an aortic-caval fistula and two weeks after surgical renal denervation (RDN). The animals in heart failure had reduced renal blood flow and a greater decrease in renal blood flow in response to Ang II, and this effect was enhanced by RDN in the heart failure group (+ fistula) but not in the hypertensive group (-fistula). The number and affinity of AT-1 receptors was similar in the groups; renal AT-2 receptors were not observed. It is concluded that the maintained level of renal AT-1 receptors, together with the enhanced response to Ang II provides the basis for the reduced RBF and renal function in HF. It is argued that since these effects are not attenuated by RDN, the beneficial effects of RDN in HF are independent of the effects on the kidney.
Main comments
The conclusion that the exaggerated renal vascular responsiveness to ANG II provides the basis for sustained influence on the renal hemodynamics is reasonable, but there is no evidence provided for the statement that it also accounts for the tubular reabsorption and development of renal dysfunction in this HF model. In addition, the assertion that the beneficial effects of RDN in HF are not mediated via renal mechanisms requires the measurement of renal function. Just because renal blood flow did not change, does not mean that renal function did not improve. Indeed, recent papers show that RDN can improve renal function in heart failure (PMID: 34333989, PMID: 31490733).
Furthermore, the lack of a measure of renal sympathetic nerve activity makes interpretation of the findings difficult. It likely that thiopental sodium will reduce RSNA and it is unclear to what extent the animals have recovered from anaesthesia and whether RSNA has returned to control levels at the time measurements were made.
There is also discussion of the role of increased plasma and intra-renal Ang II levels in this ACF TGR model but no levels are presented. If these have been measured in this model please provide citations. If they have not been measured comments regarding the role of Ang II are speculation and should be more cautious.
The finding that AT-2 receptors are not present in the kidney is referred to as an important finding (line 25), which is an exaggeration as this has been shown many times before.
It is mentioned that determining the level of Ang II receptors on a membrane fraction is a limitation. In future studies it would be worth considering using autoradiography as this would give localization of the receptors in the kidney.
It would be interesting to examine the effect of another vasopressor (e.g. phentolamine) to determine if the greater reduction in renal blood flow in the HF rats is specific to Ang II. It may be a change in the vascular muscle downstream from receptor activation.
Author Response
We appreciate the reviewer´s comments on our study and we are grateful for his encouraging and constructive suggestions.
Responses to reviewer #1:
Reviewers Comment: …….. The conclusion that the exaggerated renal vascular responsiveness to ANG II provides the basis for sustained influence on the renal hemodynamics is reasonable, but there is no evidence provided for the statement that it also accounts for the tubular reabsorption and development of renal dysfunction in this HF model. In addition, the assertion that the beneficial effects of RDN in HF are not mediated via renal mechanisms requires the measurement of renal function. Just because renal blood flow did not change, does not mean that renal function did not improve. Indeed, recent papers show that RDN can improve renal function in heart failure (PMID: 34333989, PMID: 31490733).
Authors Response:
We agree that our statement that beneficial effects of renal denervation (RDN) are not mediated by renal mechanism(s) was not fully appropriate. We agree that for such conclusion more comprehensive studies evaluating renal function are needed, such as renal micropuncture studies for addressing the effects of RDN on renal tubular function. Furthermore, to address the issue if beneficial actions of RDN are mediated by cardiac mechanism(s), studies evaluating cardiac structure and function by employing echocardiography and invasive hemodynamics are needed and this conclusion cannot be made solely on organ weights data, even if they seem convincing. Therefore, as suggested, we modified the Discussion section and we also added references to the newest studies evaluating effects of RDN on blood pressure and renal function in various experimental models (from line 485 to 499 and from 503 to 505, shown in red font, including new references, numbers 59 to 64).
Reviewers Comment: …….. Furthermore, the lack of a measure of renal sympathetic nerve activity makes interpretation of the findings difficult. It likely that thiopental sodium will reduce RSNA and it is unclear to what extent the animals have recovered from anaesthesia and whether RSNA has returned to control levels at the time measurements were made.
Authors Response:
We agree that the lack of direct measurement of renal sympathetic nerve activity (RSNA) is a limitation of our present study. Therefore, we openly discuss it in the Discussion section of the revised manuscript in the new subdivision “Limitations of the Study” (from line 539 to 547, shown in red font).
Reviewers Comment: …….. There is also discussion of the role of increased plasma and intra-renal Ang II levels in this ACF TGR model but no levels are presented. If these have been measured in this model please provide citations. If they have not been measured comments regarding the role of Ang II are speculation and should be more cautious.
Authors Response:
Thank you for this comment, we agree that this information regarding the intrarenal ANG II concentrations in this high-output model of heart failure (HF) were missing. We added this information into the Discussion section of the revised manuscript, including appropriate references (from line 429 to 433, shown in red font).
Reviewers Comment: …….. The finding that AT-2 receptors are not present in the kidney is referred to as an important finding (line 25), which is an exaggeration as this has been shown many times before.
Authors Response:
We agree and this statement was removed.
Reviewers Comment: ……. It is mentioned that determining the level of Ang II receptors on a membrane fraction is a limitation. In future studies it would be worth considering using autoradiography as this would give localization of the receptors in the kidney.
Authors Response:
We agree that such studies are needed to resolve the issue of localization and distribution of ANG II receptors in this high-output model of HF. We discuss this issue in the connection with the first limitation of our study in the Discussion section of the revised manuscript (from line 529 to 538, shown in red font).
Reviewers Comment: …….. It would be interesting to examine the effect of another vasopressor (e.g. phentolamine) to determine if the greater reduction in renal blood flow in the HF rats is specific to Ang II. It may be a change in the vascular muscle downstream from receptor activation.
Authors Response:
We agree that this is also very important issue and we discuss it in the Discussion section of the revised manuscript (from line 448 to 459, shown in red font).

Reviewer 2 Report
The authors submitted the study about the “Effects of renal denervation on the enhanced renal vascular responsiveness to angiotensin II in high-output heart failure: angiotensin II receptor binding assessment and functional studies in Ren-2 transgenic hypertensive rats”. The investigation of renal denervation has been widely studied, however, the studies regarding the heart failure model treatment approaches with renal denervation are still necessary and unclear. This article represents an important study, however below are some of the comments and concerns regarding the evaluation of this manuscript:
- Comments
1.). The authors mentioned that studies for RDN have been disregarded. We suggest the authors modify this since there are review studies on this topic (for example, the most recent one Thomas E. Sharp et al., 2021).
2.) Authors self cited their study in the introduction in reference 14, but they have a recent paper on a very related topic that might be important to introduce. Do we suggest the authors mention their previous work to clarify the meaning of this study? Findings and replication of data (if available so).
3.) Authors have published data regarding the Angiotensin II infusion in such a model in their previous paper in reference 14, but they did not delineate the meaning of this study to replicate such findings. We suggest the authors focus on either the time change approach or the need to measure the AT1 receptor. If the findings are replicated in a such way just to show the receptor expression once again this data needs to be discarded, therefore authors need to clarify this point in the hypothesis.
4.) Similarly, the RBF has been decreased after a higher dose of Angiotensin II infusion in 8 ng in the previous study (ref 14), in their model as represented clearly in previous study Figure 5.
5.) Authors mentioned a limitation of the previous study in the hypothesis to increase the emphasis in the Angiotensin Receptor 1 Density more rather than showing once again the role of Angiotensin infusion in RBF (it seems as replication of those data introduced in reference 14 as “70 days after RDN, but the same was true for all the other time points (i.e., 7, 21, 42, 56, and 84 days after RDN)”
6.) The authors did not introduce the role of AT1 receptor expression in different models with hypertension in the cortex, heart failure, etc.
7.) Similarly, the author's studies in the RDN in the treatment of HF approaches are lacking as well.
Material and Methods
1.) The authors mentioned in the method the this RDN method is fully effective, but later on raises the limitations at the end of the discussion part.
2.) Did the authors remove PVAT surrounding tissue from the groups?
3.) Why did the authors did not use either IHC or IF-related investigation in the isolated cortex to investigate the receptors (since they have previously have done mRNA expression).
4.) Why did the authors did not use a treatment group with an AT-1 receptor blocker in addition to RDN to observe the AT-1 receptor radioligand binding assay in the treatment group?
5.) We agree with their limitation of studies due to the whole cortex. We suggest the authors separate the tissues, by focusing on renal vessels and glomerular as well.
Results
1.) Why RDN did not modify the MAP in ACF but it did in non-ACF?
2.) I suggest interpreting the statistical values in the results and the current data with the real data as well rather than percentage only (include the n number for each experiment as well)
3.) Authors showed in Figure 1C that the RBF decreases produced by intrarenal ANG II were more profound in post-RDN than in intact animals, however, in the previous Figure 5 (reference 14) have shown that “RDN significantly increased the response (greater RBF decreases) in sham-operated TGR as well as ACF TGR to the higher ANG II dose (8 ng)”. Do authors need to explain this similar finding? So why this is represented in the result section (as I have mentioned in the introduction comments before).
4.) Bars in Figure 1 which stay similar in the denervated but different symbols corresponding to denervated or intact are very confusing for the reader. So please clarify this once again, separate, or either put them more clearly.
5.) In addition, the authors need to explain Figure 2 with the previous study (reference 14) Figure 4. These data are very similar so one needs to either mention any time-related difference or reason to add this data (having similar groups of observation).
6.) The authors discussed the potential limitation of their study concerning ANG II receptors in crude membrane from the renal cortex did not differ between experimental groups. However, this is very hard to explain (See PMC3023215). We agree with the reduction of AT-2 receptors but this needs to be studied in more detail.
7.) The authors need to divide the AT-1 receptor and the AT-2 receptor itself. In addition, authors might focus on other Angiotensin II related receptors such as Angiotensin II metabolite Angiotensin I-7 in Mas Receptor. In addition, there is Angiotensin receptor 4 as well. So such binding might be non-selective in terms of appropriate quantification in such a manner by blocking the AT-2 receptor only “The AT1 receptor binding was defined as the difference between nonspecific binding minus binding in the presence of PD 123319, the AT2 receptor blocker”.
Discussion
1.) Authors need to discuss the role of RDN related studies in heart failure and the positive impact itself rather than moving to increased vascular responsiveness possibly further augmented by RDN. We suggest the authors use the term renal blood flow rather than vascular responsiveness since this term is used more for measurement of the direct vascular responsiveness through in vivo or in vitro studies.
2.) I agree with the decrease of AT-2 receptors, and such counteraction or cross-talk were not observed in this study.
3.) Authors need to discuss the consistent studies which showed that the AT-1 receptor plays role in heart failure which is recovered by RDN. However, in such a model or methodological approach, this is not being observed.
4.) Authors discussed the exclusive role of the AT-1 receptor in the related models by using the word in previous “this is in agreement”. Again this might compromise such findings since your data do not see any change in AT-1 receptor with such methodological investigation, therefore authors need to improve this part in a way that they distinct from previous approaches and findings “this is contradictory to what has been seen so far in other models”
5.) Authors tried to minimize the role of AT-2 receptor in hypertension “This suggests that AT2 receptors do not play an important role in the regulation of ANG II production and blood pressure in this model of ANG II-dependent hypertension” and also heart failure (for example, PMID: 10642292; PMID: 28943106). Authors need to be careful that such treatment might have an impact since they do not use any kind of treatment approaches with either ACE, ARBs, or AT-2 selective agonists in such model.
6.) Authors discussed their treatment approach “Therefore, our present findings, along with our recent evidence that selective intrarenal AT1 receptor blockade elicits greater increases in RBF than in intact sham-operated TGR [14], are consistent with the concept that in the high-output HF model renal functional derangements are strongly ANG II-dependent” This is inconsistent with what is seen in this study, meaning that Angiotensin receptors are similar in all the groups and the RBF was increased more, it does not make sense? We would at least expect the increase in AT-1 receptors with such a decrease of RBF.
7.) Again we agree with the authors' limitation of their study, which is still the high number of the limitation and things to do to better clarify this issue. This is in terms of the RDN model and as well the approach of targeting the AT-1 receptor. I would add the pharmacological investigations by using the in vivo treatment approaches
Author Response
We appreciate the reviewer´s comments on our study and we are grateful for his encouraging and constructive suggestions.
Responses to reviewer #2:
Reviewers Comment: …….. The authors mentioned that studies for RDN have been disregarded. We suggest the authors modify this since there are review studies on this topic (for example, the most recent one Thomas E. Sharp et al., 2021).
Authors Response: We agree that our statement was not quite correct, hence we modified it in the revised version of the manuscript as suggested (from line 56 to 66, shown in red font). We agree that the newest findings from experimental and clinical studies have considerably changed the view on the RDN and it is now accepted not only as a safe method, but as a potential new approach for the treatment of various cardiovascular diseases.
Reviewers Comment: …….. Authors self cited their study in the introduction in reference 14, but they have a recent paper on a very related topic that might be important to introduce. Do we suggest the authors mention their previous work to clarify the meaning of this study? Findings and replication of data (if available so).
Authors Response:
We agree and therefore, we introduced and discussed our findings on the effects of RDN in fawn hooded hypertensive (FHH) rats in the Introduction section (from line 66 to 76, shown in red font). We agree that recent findings from the above mentioned paper are important and further support the need for and the rationale of the present study.
Reviewers Comment: …….. Authors have published data regarding the Angiotensin II infusion in such a model in their previous paper in reference 14, but they did not delineate the meaning of this study to replicate such findings. We suggest the authors focus on either the time change approach or the need to measure the AT1 receptor. If the findings are replicated in a such way just to show the receptor expression once again this data needs to be discarded, therefore authors need to clarify this point in the hypothesis.
Authors Response: We agree that it was not clearly explained why we repeated certain experiments, i.e. the renal blood flow (RBF) responsiveness to intrarenal bolus administration of ANG II. Therefore, in the Introduction section of the revised manuscript, we provide more detailed justification for such experiments (from line 104 to 111, shown in red font). The present study is an obvious continuation of the prior experiments and inspired by previous results, which were not fully anticipated. The aim of the current study was to obtain a deeper insight in the role of exaggerated RBF responsiveness to ANG II, hence we wanted to make sure that the functional/physiological experiments (RBF responsiveness) and biochemical evaluation (binding studies) agree with each other when performed in the same animals and the same biological material.
Reviewers Comment: …….. Similarly, the RBF has been decreased after a higher dose of Angiotensin II infusion in 8 ng in the previous study (ref 14), in their model as represented clearly in previous study Figure 5.
Authors Response: This is principally the same concern and therefore please see our response above.
Reviewers Comment: …….. Authors mentioned a limitation of the previous study in the hypothesis to increase the emphasis in the Angiotensin Receptor 1 Density more rather than showing once again the role of Angiotensin infusion in RBF (it seems as replication of those data introduced in reference 14 as “70 days after RDN, but the same was true for all the other time points (i.e., 7, 21, 42, 56, and 84 days after RDN)”
Authors Response:
After careful revision of the Introduction section we admit that our original formulation of the rationale was a bit perplexing, hence the main message may have been lost. The present study, was focused on the characterization of the kidney ANG II receptors with particular focus on the role of AT2 receptors subtype. Therefore, we modified the last paragraph in the Introduction section (from line 97 to 111, shown in red font) and also highlighted our hypothesis at the beginning of the Discussion section of the revised manuscript (from line 359 to 371). We hope that it is now clear and understandable to readers.
Reviewers Comment: …….. The authors did not introduce the role of AT1 receptor expression in different models with hypertension in the cortex, heart failure, etc.
Authors Response: We agree and therefore we discuss the role of AT1 and AT2 expressions (and also of other components of RAS) in the HF in the Discussion section (from line 363 to 371, shown in red font). However, we decided not to discuss this issue in various models of experimental hypertension, because the results are conflicting and depend on the model that was employed (e.g. Brown et al. J Mol Cell Cardiol. 29: 2925-2929, 1997). We believe that the Discussion (and consequently the whole manuscript, which is already fairly long) would become inappropriately lengthy, hence the main “message” of our study could be even more difficult to comprehend. Nevertheless, we agree that this is a very important issue and we plan to address it in our future studies.
Reviewers Comment: …….. Similarly, the author's studies in the RDN in the treatment of HF approaches are lacking as well.
Authors Response: We modified our Introduction section as suggested – please see the response to the first comment.
Reviewers Comment: …….. The authors mentioned in the method the this RDN method is fully effective, but later on raises the limitations at the end of the discussion part.
Authors Response: The limitation which was mentioned in the end of Discussion section regards another technique of RDN, i.e. the selective ablation of renal afferent nerves. The method which we used in this study (mechanical+10% phenol solution) causes ablation of both afferent and efferent renal nerves and it is long lasting and well described in literature, including our own studies. We modified appropriate paragraphs in the Methods section (from line 154 to 156, shown in red font) to make it more understandable for the readers.
Reviewers Comment: …….. Did the authors remove PVAT surrounding tissue from the groups?
Authors Response: Yes we have. We added this information to the Methods section (from line 139 to 140, shown in red font).
Reviewers Comment: …….. Why did the authors did not use either IHC or IF-related investigation in the isolated cortex to investigate the receptors (since they have previously have done mRNA expression).
Authors Response: We recognize that this is a limitation of our present study and we openly discuss it in the new section “Limitation of the Study” as the first limitation (from line 529 to 537). We are fully aware that such studies are needed in future.
Reviewers Comment: …….. Why did the authors did not use a treatment group with an AT-1 receptor blocker in addition to RDN to observe the AT-1 receptor radioligand binding assay in the treatment group?
Authors Response: Our study was mainly focused on the role of AT2 receptors, because their role is by many investigators (including our own laboratory) disregarded and usually the main focus is on the role of AT1 receptors (there is no doubt that AT1 receptors dominantly mediate physiological and pathophysiological actions of ANG II). For technical reasons, when binding studies are performed in the presence of AT1 receptor antagonist, such as losartan, it is almost impossible to define the role of AT2 receptors. At higher doses, losartan, alleggedly selective, binds also to AT2 receptors. Therefore, in the present study we decided to perform the experiments only in the presence of the selective AT2 receptor antagonist. In the revised version of the manuscript we tried to clarify this issue and put more emphasis on the main hypothesis, which was focused on the role of AT2 receptors (from line 97 to 101 and from line 359 to 371, shown in red font) and we believe that it is now more understandable to the readers.
Reviewers Comment: …….. We agree with their limitation of studies due to the whole cortex. We suggest the authors separate the tissues, by focusing on renal vessels and glomerular as well.
Authors Response:
We agree and in our future studies we will separate the tissues. We openly admit this limitation in the revised version of the manuscript (from line 529 to 538).
Reviewers Comment: …….. Why RDN did not modify the MAP in ACF but it did in non-ACF?
Authors Response:
We are very pleased that reviewer noticed this important finding, however we can provide no adequate explanation. We discuss this issue as the fifth important finding of our present study and it is obvious that future studies are needed to address it (from line 522 to 526, shown in red font).
Reviewers Comment: …….. I suggest interpreting the statistical values in the results and the current data with the real data as well rather than percentage only (include the n number for each experiment as well)
Authors Response:
We agree and the results from the Series 1 are now presented in absolute values, but we also retained the graphs with the percentage changes to make it better readable. We prepared a new Figure 1 and the whole Results section was rearranged (from line 276 to 320, shown in red font). The statistical analysis was performed on absolute values.
Reviewers Comment: …….. Authors showed in Figure 1C that the RBF decreases produced by intrarenal ANG II were more profound in post-RDN than in intact animals, however, in the previous Figure 5 (reference 14) have shown that “RDN significantly increased the response (greater RBF decreases) in sham-operated TGR as well as ACF TGR to the higher ANG II dose (8 ng)”. Do authors need to explain this similar finding? So why this is represented in the result section (as I have mentioned in the introduction comments before).
Authors Response:
We believe that this concern was already raised before and therefore please see our previous responses (especially to the third comment). We hope that we have clarified this issue.
Reviewers Comment: …….. Bars in Figure 1 which stay similar in the denervated but different symbols corresponding to denervated or intact are very confusing for the reader. So please clarify this once again, separate, or either put them more clearly.
Authors Response:
We agree and therefore, we prepared new figures and we believe that graphs are now clearer and understandable.
Reviewers Comment: …….. In addition, the authors need to explain Figure 2 with the previous study (reference 14) Figure 4. These data are very similar so one needs to either mention any time-related difference or reason to add this data (having similar groups of observation).
Authors Response:
We believe that this concern was already raised before and therefore please see our previous responses (especially to the third comment). We hope to have clarified this issue.
Reviewers Comment: …….. The authors discussed the potential limitation of their study concerning ANG II receptors in crude membrane from the renal cortex did not differ between experimental groups. However, this is very hard to explain (See PMC3023215). We agree with the reduction of AT-2 receptors but this needs to be studied in more detail.
Authors Response:
Thank you for this comment. The above mentioned study (Clayton et al., reference 51 in our revised manuscript) further underscores the potential role of AT2 receptors and particularly the effects of RDN on ANG II receptors expression in the kidney. We cannot provide fully satisfactory explanation for this divergence between our and their findings. We discuss this issue in the revised version of the manuscript (from line 391 to 408, shown in red font) and we believe that we have addressed it appropriately.
Reviewers Comment: …….. The authors need to divide the AT-1 receptor and the AT-2 receptor itself. In addition, authors might focus on other Angiotensin II related receptors such as Angiotensin II metabolite Angiotensin I-7 in Mas Receptor. In addition, there is Angiotensin receptor 4 as well. So such binding might be non-selective in terms of appropriate quantification in such a manner by blocking the AT-2 receptor only “The AT1 receptor binding was defined as the difference between nonspecific binding minus binding in the presence of PD 123319, the AT2 receptor blocker”.
Authors Response:
Thank you for this comment, partially, we addressed this issue earlier (please see the answer to yours comment above). We agree that there is definitely more to do and we are considering to investigate other receptors (and in general more RAS components) in future studies. We address this issue in the revised version of the manuscript (from line 375 to 382 and from line 529 to 538, shown in red font).
Reviewers Comment: …….. Authors need to discuss the role of RDN related studies in heart failure and the positive impact itself rather than moving to increased vascular responsiveness possibly further augmented by RDN. We suggest the authors use the term renal blood flow rather than vascular responsiveness since this term is used more for measurement of the direct vascular responsiveness through in vivo or in vitro studies.
Authors Response:
Thank you for this comment. We believe that this issue is now appropriately addressed in the Introduction and Discussion sections of the revised manuscript (please see all changes shown in red font, we use the term “RBF responsiveness”).
Reviewers Comment: …….. I agree with the decrease of AT-2 receptors, and such counteraction or cross-talk were not observed in this study.
Authors Response:
We also agree.
Reviewers Comment: …….. Authors need to discuss the consistent studies which showed that the AT-1 receptor plays role in heart failure which is recovered by RDN. However, in such a model or methodological approach, this is not being observed.
Authors Response:
We decided not to discuss this issue, because it is far beyond the scope of the present study and from our point of view such discussion would only distract readers from the main message (from our point of view the Discussion has already reached maximal length). Nevertheless, we agree this is important issue, but deserves separate studies.
Reviewers Comment: …….. Authors discussed the exclusive role of the AT-1 receptor in the related models by using the word in previous “this is in agreement”. Again this might compromise such findings since your data do not see any change in AT-1 receptor with such methodological investigation, therefore authors need to improve this part in a way that they distinct from previous approaches and findings “this is contradictory to what has been seen so far in other models”
Authors Response:
As suggested, we modified this statement in the Discussion section of the revised manuscript (from line 374 to 382, shown in red font).
Reviewers Comment: …….. Authors tried to minimize the role of AT-2 receptor in hypertension “This suggests that AT2 receptors do not play an important role in the regulation of ANG II production and blood pressure in this model of ANG II-dependent hypertension” and also heart failure (for example, PMID: 10642292; PMID: 28943106). Authors need to be careful that such treatment might have an impact since they do not use any kind of treatment approaches with either ACE, ARBs, or AT-2 selective agonists in such model.
Authors Response:
We agree and therefore we modified our statement in the Discussion section of the revised manuscript (from line 391 to 408, shown in red font).
Reviewers Comment: …….. Authors discussed their treatment approach “Therefore, our present findings, along with our recent evidence that selective intrarenal AT1 receptor blockade elicits greater increases in RBF than in intact sham-operated TGR [14], are consistent with the concept that in the high-output HF model renal functional derangements are strongly ANG II-dependent” This is inconsistent with what is seen in this study, meaning that Angiotensin receptors are similar in all the groups and the RBF was increased more, it does not make sense? We would at least expect the increase in AT-1 receptors with such a decrease of RBF.
Authors Response:
We have modified paragraphs before and after this statement and we believe that it is now better understandable to readers. As discussed in the whole section addressing the second important finding or our study (from line 409 to 433), one would expect that increased intrarenal ANG II concentration should downregulate the expression of AT1 receptors, but it did not. Therefore, similar expression of AT1 receptors as observed in sham-operated animals, can be considered as inappropriately high (relative to the increase in AT1 receptors in the kidneys of ACF animals). In addition, we previously found that selective intrarenal blockade of AT1 receptors (intrarenal administration of losartan) causes substantially higher increase in RBF and sodium excretion in ACF TGR than in sham-operated TGR. Moreover, considering also the exaggerated RBF decrease after intrarenal ANG II administration (as observed in our present and previous study), evidence is provided that kidneys of ACF TGR are under sustained (in this case negative) influence of increased intrarenal ANG II levels. We hope that the introduced modifications to this part of the Discussion section provide now more comprehensive explanation to the readers.
Reviewers Comment: …….. Again we agree with the authors' limitation of their study, which is still the high number of the limitation and things to do to better clarify this issue. This is in terms of the RDN model and as well the approach of targeting the AT-1 receptor. I would add the pharmacological investigations by using the in vivo treatment approaches
Authors Response:
We are fully aware of many limitations of our present study. We believe that we did our best to address them openly and in detail. However, even despite all these limitations we are convinced that our present study provides important information on the pathophysiology of cardiorenal syndrome (“several pieces to very complicated puzzle”).

Reviewer 3 Report
Review
In the submitted paper the authors prove in a very complex in vivo study that in Ren-2 transgenic hypertensive rats (1) in which heart failure has been induced by aortocaval fistula (2), the kidneys denervated (3) sensitivity of renal blood flow to intrarenal Ang2 infusion (4) increases which is not the result of appearance of angiotensin II type 2 receptors or overall elevation of the expression of Ang II type 1 receptors in renal cortical tissue. An other clinically important observation was that renal denervation extended the compensated state of heart failure after aorto-caval shunt.
The control of extracellular fluid volume and composition in hypertension and in heart failure associated with hypertension are very important clinical questions, the mechanism of several effective clinical treatments is still not fully understood, any new idea will be highly esteemed.
The problem with the paper is the complexity of the interventions (see the numbering in parentheses above), having multiple branching of their action, even with potentially intermingling pathomechanisms. The supposed lines of action do exist, but together with several additional effects which were not taken into consideration.
- All experiments were performed on Ren-2 transgenic rats. Such rodent strain has been constructed in 1990 by Mullins JJ et al. (Nature 1990;344:541) and Langheinrich M et al. (Am J Hypert 1996;9:506) and reached some limited popularity among specialists. An additional mouse renin gene is inserted into the rat genome, the result is massive elevation of blood pressure, the mechanism of which is still not fully understood (Rong P et al Hypertension 2003;42:523, Seccia TM et al. J Hypertension 2008;26:2022). We still do not know in what tissues and how the expression of the inserted gene will be controlled. This fact should command certain caution at interpretation of results. Introduction of analog experiments on non-transgenic wild animals could have added some safety for conclusions.
- The aortocaval fistula is also an accepted but not too popular model of heart failure. The increased venous filling will induce elevated cardiac output and later heart failure due to exhaustion, but the elevation of systemic venous pressure will be higher than in cases of heart failure of similar degree. Elevated venous pressure will have then extensive kidney, liver, etc. effects. The function of low-pressure baroreceptors that have important role in reflex control of juxtaglomerular function will be much altered. No attempts have been made to outline these effects (Eg. measuring plasma and urine lab values, water balance, glomerular filtration rate, etc).
- Kidney denervation is now a last resort intervention to treat malign, resistant hypertension, few doctors will advise it. The method the authors apply can harm the renal artery wall; any stricture of the renal artery due to phenol treatment can induce hypertension. Even intrarenal thrombosis due to endothelial damage can not be excluded.
- “Kidney infusion of AngII” is not fully clear, it can induce local kidney necrosis, the injected drug will get back into the circulation inducing systemic effects.
- One cannot expect a homogenous distribution of Ang II type 1 and type 2 receptors in the very different vascular and tubular segments of the kidney. Immuno-histochemistry with commercially available antibodies would be more appropriate to study their expression and tissue distribution in different conditions.
- Their statement that the beneficial effect of renal denervation observed in hypertensive heart failure can be explained by nonrenal mechanisms seems to be not substantially founded. Of the potential effects of cessation of sympathetic outflow to the kidneys only the expression of Ang II receptors, renal blood flow and their alterations in response to intrarenal Ang II were studied.
These limitations should be introduced into the revised form of the paper.
Author Response
We appreciate the reviewer´s comments on our study and we are grateful for his encouraging and constructive suggestions.
Responses to reviewer #3:
Reviewers Comment: …….. All experiments were performed on Ren-2 transgenic rats. Such rodent strain has been constructed in 1990 by Mullins JJ et al. (Nature 1990;344:541) and Langheinrich M et al. (Am J Hypert 1996;9:506) and reached some limited popularity among specialists. An additional mouse renin gene is inserted into the rat genome, the result is massive elevation of blood pressure, the mechanism of which is still not fully understood (Rong P et al Hypertension 2003;42:523, Seccia TM et al. J Hypertension 2008;26:2022). We still do not know in what tissues and how the expression of the inserted gene will be controlled. This fact should command certain caution at interpretation of results. Introduction of analog experiments on non-transgenic wild animals could have added some safety for conclusions.
Authors Response:
We agree with this comment and we are fully aware of TGR shortcomings. In the revised version of the manuscript we discuss the said shortcomings in a new section labeled “Limitations of the Study” (from line 548 to 563, shown in red font).
Reviewers Comment: …….. The aortocaval fistula is also an accepted but not too popular model of heart failure. The increased venous filling will induce elevated cardiac output and later heart failure due to exhaustion, but the elevation of systemic venous pressure will be higher than in cases of heart failure of similar degree. Elevated venous pressure will have then extensive kidney, liver, etc. effects. The function of low-pressure baroreceptors that have important role in reflex control of juxtaglomerular function will be much altered. No attempts have been made to outline these effects (Eg. measuring plasma and urine lab values, water balance, glomerular filtration rate, etc).
Authors Response:
Again, we agree and similarly as above with the TGR strain, the limitation of ACF model is discussed as the fourth limitation of our present study (form line 564 to 576, shown in red font). However, we employ the ACF model in our laboratory for more than ten years now and we believe that we have sufficiently characterized the basal values in various rat strains, also in TGR (please see, for example: Kala et al. Physiol Res. 2018;67(3): 401–415; Honetschlagerová et al. Clin Exp Hypertens. 2021, 43(6):522-535; Vacková et al. Kidney Blood Press Res. 2019;44(4):792-809; Vacková et al. Front Pharmacol. 2019 Jan 23;10:18;), therefore we do not believe that repetition of these measurements would be needed.
Reviewers Comment: …….. Kidney denervation is now a last resort intervention to treat malign, resistant hypertension, few doctors will advise it. The method the authors apply can harm the renal artery wall; any stricture of the renal artery due to phenol treatment can induce hypertension. Even intrarenal thrombosis due to endothelial damage can not be excluded.
Authors Response:
We understand this concern and fully agree that the chosen method of RDN is burdened with some potential harmful effects. It is true, that the combination of surgical and chemical approach (particularly the use of phenol solution) can cause some damage of renal arteries and we observe it in some cases. However, such animals are always excluded (even if we have only a slight doubt about the potential damage). In order to clarify this issue, in the revised version of the manuscript in the Methods section we provided more detailed description of RDN technique and added the exclusion criteria, including the exact number of animals that were excluded (from line 144 to 150, shown in red font).
Regarding the first comment, that only few doctors will advise renal denervation to treat hypertension: we do not fully agree. It is true that over the years there was a lot of confusion and conflicting reports about this treatment option, however based on the newest results of several clinical trials, we believe that it represents an alternative/additional, but valuable treatment strategy (please see Schmieder et al. European Society of Hypertension position paper on renal denervation 2021. J Hypertens. 2021; 39: 1733-1741.). We cite this new information in the Introduction section of the revised manuscript and we are deeply convinced, that not only we (basic researchers), but, at present, also the physicians that are involved in the treatment of patients with so called resistant hypertension, have concluded that RDN could be new treatment strategy for such patients (of course with catheter-based RDN technique etc.), including HF patients.
Reviewers Comment: …….. “Kidney infusion of AngII” is not fully clear, it can induce local kidney necrosis, the injected drug will get back into the circulation inducing systemic effects.
Authors Response:
We agree that the description of intrarenal administration of vasoactive drugs (in this case ANG II) was not clear and even to some degree misleading. Therefore, to the Methods section we added more detailed description of this technique, including appropriate original references, and we believe that this technical (but very important) issue will be now clear to the readers (from line 186 to 201, shown in red font).
Reviewers Comment: …….. One cannot expect a homogenous distribution of Ang II type 1 and type 2 receptors in the very different vascular and tubular segments of the kidney. Immuno-histochemistry with commercially available antibodies would be more appropriate to study their expression and tissue distribution in different conditions.
Authors Response:
We agree and we are aware that this is a limitation of our study. Therefore, we added the special section entitled “Limitations of the Study” and this issue is discussed as the “first limitation” (from line 529 to 538, shown in red font).
Reviewers Comment: …….. Their statement that the beneficial effect of renal denervation observed in hypertensive heart failure can be explained by nonrenal mechanisms seems to be not substantially founded. Of the potential effects of cessation of sympathetic outflow to the kidneys only the expression of Ang II receptors, renal blood flow and their alterations in response to intrarenal Ang II were studied.
Authors Response:
We agree that our original statement that the beneficial effects of RDN are unlikely to be mediated by renal mechanism(s) was grossly exaggerated and not supported by our present findings. The same concern was raised by Reviewer #1 and we admit that more comprehensive studies evaluating renal function are needed for such conclusion to be valid. To address the effects of RDN on renal tubular function, renal micropuncture studies would be required. Furthermore, to address the issue if beneficial actions of RDN are primarely mediated by cardiac mechanism(s), studies evaluating cardiac structure and function by employing echocardiography and invasive hemodynamics are needed. We admit that the above conclusion cannot be made solely on organ weights data, even if they appear convincing. Therefore, as suggested by the Referees, we appropriately modified the Discussion section and we also added references of newest studies evaluating effects of RDN on blood pressure and renal function in various experimental models (from line 485 to 499 and from 503 to 505, shown in red font, including new references numbers 59 to 64).
Round 2
Reviewer 3 Report
The paper much improved as a result of implemented alterations.
The Introduction gives a wider approach to the topic and together with the the Limitations section of the Discussion they provide a better orientation for the readers. The paper is now a reliable surway of the role of AngII in the kidney denervation effects.